# Advanced Computation Capacity Modeling for Delay-Constrained Placement of IoT Services

**DOI:** 10.3390/s20143830

**Published:** 2020-07-09

**Authors:** Balázs Németh, Balázs Sonkoly

**Affiliations:** MTA-BME Network Softwarization Research Group, Budapest University of Technology and Economics, 1111 Budapest, Hungary; sonkoly@tmit.bme.hu

**Keywords:** network abstraction, virtual network embedding, 5G infrastructure, IoT services, Kubernetes

## Abstract

A vast range of sensors gather data about our environment, industries and homes. The great profit hidden in this data can only be exploited if it is integrated with relevant services for analysis and usage. A core concept of the Internet of Things targets this business opportunity through various applications. The virtualized and software-controlled 5G networks are expected to achieve the scale and dynamicity of communication networks required by Internet of Things (IoT). As the computation and communication infrastructure rapidly evolves, the corresponding substrate models of service placement algorithms lag behind, failing to appropriately describe resource abstraction and dynamic features. Our paper provides an extension to existing IoT service placement algorithms to enable them to keep up with the latest infrastructure evolution, while maintaining their existing attributes, such as end-to-end delay constraints and the cost minimization objective. We complement our recent work on 5G service placement algorithms by theoretical foundation for resource abstraction, elasticity and delay constraint. We propose efficient solutions for the problems of aggregating computation resource capacities and behavior prediction of dynamic Kubernetes infrastructure in a delay-constrained service embedding framework. Our results are supported by mathematical theorems whose proofs are presented in detail.

## 1. Introduction

Digital applications relying on information gathered by sensors and transmitted by infocommunication networks are becoming pervasive across industries. Data are continuously collected from production lines in mass-producing factories, while warehousing robots, delivery drones and driverless vehicles are ensuring the autonomous, cost efficient logistics of getting products to the end-customers. Agricultural sensors are providing unprecedented insight into the efficiency of industrial-scale food and crops production. A wide range of smart city sensors and actuators for traffic control, parking optimization, public transportation and sharing economy based urban mobility solutions make metropolitan lives more convenient and environment friendly. Autonomous video surveillance, consumer intelligence based advertisements, smart home appliances and smart phones make our lives safer, comfortable and filled with personalized user experience. Enormous amount of information and optimization opportunities lurk in this massive, hyper-connected system. The value of this huge amount of data can only be exploited if it is collected, analyzed, understood and acted upon. That is the core process to monetize information collected by various types of sensors, and that is the essence of the Internet of Things (IoT).

The high service quality requirements driven by consumer demands of the information society, pushes communication technology innovation and development forward. New applications built on top of the vast amount of process intelligence are coming out rapidly, and innovation is ongoing with a never-seen-before pace. These demands of the current world characterize the use-cases of 5G networks and their functional requirements, such as massive Machine Type Communication (mMTC), Ultra-Reliable Low-Latency Communication (URLLC) and enhanced Mobile Broadband (eMBB) [1]. A recent surge of 5G prototype architectures and the ongoing competition for commercial 5G network deployments are academia’s and industry’s response to these challenging requirements. 5G systems utilize the paradigms of Network Function Virtualization (NFV) and Software-Defined Networking (SDN) to decouple the services’ business logic from the hardware, and realize an efficient management of computation and communication resources [2].

A central optimization problem in realizing IoT applications using the enabling 5G technology, is allocating service components to resources, which is formalized in multiple NP-hard variants of the service placement problem. Service placement is one of the most difficult problems, as no solution exists which is scalable (i.e., do not rely on directly solving integer programs), provably meets all constraints and provides performance guarantees [3,4]. Due to the immediate relevance of the service placement problem, a wide range of practical heuristic algorithms have emerged, which are easy-to-deploy, highly extensible and possible to be continuously developed in an actual 5G software architecture [5].

In parallel to algorithmic research on the service placement problem, the computation infrastructure is also evolving, becoming more dynamic and heterogeneous to contribute to meeting the challenging, service-level user requirements. In contrast, the infrastructure model of placement algorithms mostly consider only fixed substrate capacities and rarely goes beyond different capacity volumes to model heterogeneity. In addition to tackling heterogeneity and dynamicity, infrastructure modeling is an essential tool to handle the difficulty of the VNE problem by devising higher abstraction resource views for better algorithmic scalabilty.

In this paper, we contribute to 5G-enabled IoT service placement systems by proposing theoretically founded infrastructure abstractions for aggregating capacities and modeling elastic resources. We propose a service component placement heuristic algorithm design schema for meeting constraints on service path delay requirements. We show how these results can be integrated into proven prototype architectures. The focus of our paper is modeling the modern capabilities of distributed computing infrastructures, where services are deployed with respecting end-to-end requirements. More specifically, our contribution is threefold:We propose a method to design the step-by-step delay bounds of a greedy service placement algorithm to meet end-to-end delay requirements.We define capacity aggregation to efficiently abstract the network resources, targeting scalability and information hiding. We propose an approximation algorithm to create the aggregate infrastructure view with the desired abstraction level.We propose an elastic resource model and admission control for the Kubernetes container management system’s Horizontal Pod Autoscaler feature [6,7], and show how it can be integrated with service placement algorithms.

All of our results are complemented with rigorous mathematical proofs, shown in the Appendix A, Appendix B and Appendix C.

The paper is structured as follows. Section 2 puts our work in the context of a concrete IoT service deployment, and reviews related work on network abstraction and service placement. Section 3 explains the base system model, and introduces the service placement problem variant solved by our earlier works. Section 4 presents our results on end-to-end delay constraints and integrates it into our service placement framework. Our theoretical results on infrastructure modeling are presented in Section 5 and Section 6, which incrementally extend our base system model with capacity aggregation capability and resource elasticity modeling. Integration of our resource models to 5G-enabeld IoT orchestration systems is presented in Section 7. Finally, Section 8 concludes the paper.

## 2. Background and Related Work

### 2.1. Background

Meeting the Internet of Things communication requirements such as connectivity of vast number of devices with high mobility, low latency and high reliability bring grand challenges to communication networks. These are addressed by the 5G mobile technology, which enables the monetization of various sensor-aided IoT services. The enabling relationship between 5G and IoT is examined in [1], where the state-of-the-art is extensively surveyed, gaps and research challenges are identified. The survey studies Mobile Edge Computing (MEC), data center performance optimization, edge cloud and virtualization technologies to identify their role in realizing 5G-enabled IoT applications, such as cloud robotics, industrial cyber physical systems, smart vehicles and tactile internet. Facilitating the interactions between various entities of the networking ecosystem is required to realize end-to-end IoT services. Authors of [8] present their prototype for a functional 5G architecture for optimizing end-to-end IoT network slices, which is integrated with the Kubernetes-based Google Compute Engine.

In the following we present a remote security camera analytics and alerts use case, where the computation intensive and delay sensitive IoT services are deployed across big geographic distances by an Over-the-Top Service Provider (OTT SP). Figure 1 shows a wide area network from the Internet Topology Zoo [9], where a Communication Service Provider (CSP) gives network connection for an OTT SP between its service sites. The location of the security cameras is to the west (loi_w), while the end-user location ($sap_{e}$) is to the east. Service Access Points (SAPs) are the user locations where the service is to be consumed, possibly by another service or a person. Location of Interests (LOIs) represent location-bound points, where the sensors (i.e., security cameras) are located. These should be used as service delay reference points, because the service quality is primarily experienced here. All auxiliary functions realize the service needs to be allocated on a network path between the reference points with meeting the strict delay requirement. The CSP provides communication capabilities between the distributed cloud of a Cloud Provider (CP) having its computation nodes apart with significant geographic distance. The OTT SP is in a business relationship with a CP, reaching its data center through the CSP’s network, while also owning a private cloud proximate to site loi_w.

If the OTT SP wishes to provide high quality IoT service over the rented infrastructure, it needs to know information about the CSPs network in significant detail to make informed decision about the IoT service component placements. On the other hand, the CSP may want to hide information from its tenants as they might be valuable or sensitive [10], where appropriate network and computation resource abstractions are essential. In addition, the CSP might want to optimize the running services’ resource allocation to achieve higher computation resource efficiency, which should not be perceived by the OTT SP on its rented infrastructure view. The OTT SP’s orchestration system needs to work on an appropriate infrastructure view which resolves the mentioned conflicts. Aggregating servers into higher abstraction nodes (as shown in CSP Aggregate node of Figure 1) also helps the embedding algorithm to scale up to carrier-grade network sizes. Finally, the Cloud Provider should also be able to give a descriptive view of its Kubernetes-based, optimized, dynamic data center resources.

### 2.2. Network Abstraction

Network infrastructure abstraction has always been important in modern network management to address the various application needs serving on top of the increasingly complex communication substrate. In a simple case, when only throughput requirements between endpoints need to be satisfied, the hose model has been proposed to describe Virtual Private Networks (VPNs) over a common substrate. The hose model has proved to be useful to efficiently provision the VPNs using a convenient tree structure [11]. In addition to meeting the bandwidth requirements, routing costs must be considered in practical cases. Such a system has been described by an abstracted network map [12], where the significant nodes are connected by cheapest, i.e., shortest in terms of cost metric, and highest bandwidth capacity paths, i.e., widest paths in terms of throughput. As virtual networks are the most valuable when they connect geographically distant locations, the underlying network infrastructure is likely to be managed by multiple providers. Virtual Network Embedding (VNE) is studied across multiple administrative domains in [10], where the authors devise a distributed algorithm, segmenting the virtual networks and embedding their routes in individual steps. The presented model lacks admission control capabilities on the domain capacity resources, which is one of the main contribution of our work.

As Software Defined Networking (SDN) enabled the network-wide control from a logically centralized controller, end-to-end network management based on high abstraction policies is convenient. SDN control architecture proves to be indispensable to implement IoT networks, furthermore, their appropriate performance requires distributed SDN controller placement [13], where abstracting the low-level management of individual devices is important. Network infrastructure abstraction is a key component to make the SDN concepts reach their full potential as studied in an SDN scalability survey [14]. Addressing these issues, the Big Switch model has been proposed to describe a network by its transport characteristics between the endpoints [15], which can be used to provision user traffic on top of the network, without studying its low-level details. Thanks to Network Function Virtualization (NFV), network services are decoupled from specialized hardware, packaged using virtualization technologies (such as containers and micro-services) and freely moved across computation locations [16]. Our recent works have generalized the Big Switch model to include computation capabilities of a network and keep up infrastructure modeling with the increasingly interleaving computation and communication resources. We have defined the Big Switch-Big Software (BiS-BiS) model [17], which has been used to build a multi-domain SDN/NFV-aware orchestration system [18,19]. These earlier versions of the BiS-BiS model naively sum the computation capacities and aggregate the functional capabilities of the hidden infrastructure, which ignores essential bottlenecks and lacks configuration options for the abstraction-performance trade-off. Emerging deployments of the NFV/SDN-enabled technologies in Mobile Edge Computing (MEC), Fog Computing, Industry 4.0, Internet of Things (IoT) and 5G make the networks increasingly heterogeneous. Thus, the naive BiS-BiS model’s shortcomings are crucial in real deployments. Our current work addresses mathematically founded description of abstract, highly distributed, dynamic compute and network infrastructures.

### 2.3. Service Placement

The central problem in SDN and NFV-based computation network architectures is the service component placement problem, which has been well-studied by both industry and academia, stated in many forms. The general problem of service placement benefits from multiple areas of network management research, such as the Virtual Network Embedding (VNE) problem [20], Virtual Network Function (VNF) placement or Service Function Chain (SFC) resource allocation problem. Relations between these problem variants, and mutually useful results have been studied by survey papers [21]. The core of the service component placement problem addresses how to place the service-composing VNFs and how to route their connections with various constraints on the communication and computation characteristics. The problem is notoriously NP-hard and very difficult to approximate [3]. Moreover, modeling dynamic infrastructure capabilities such as reactive, automatic scale out of service components managed by modern container management systems is challenging. These behaviors cannot be formalized using linear constraints for integer programs, making solver-based optimization approaches hopeless [22]. A wide range of heuristic approaches have been proposed to solve many variants of the service placement problem, which are much more promising to meet high quality service requirements scaling to the immense size of IoT networks. We target complementing heuristic placement algorithms by providing theoretically founded generalizations of the infrastructure models to describe elasticity and abstraction. Indicating the hardness of the service placement related problems, no scalable solutions have been proposed so far which have performance guarantees and provably do not violate capacity constraints [4].

The surveys in [20,21] extensively study and categorize the earlier and more recent papers of VNE, defining a general optimization model, and proposing a taxonomy on the solutions based on their modeled features, algorithmic approaches and considered constraints. Focusing more on the technological aspects of VNE related problems, taking into account the modern developments in virtualization technologies, such as containerization, the survey in [5] presents a novel classification.

Concluding from the above state-of-the-art studies, many heuristics use a greedy approach to target easy prototypization and extension capabilities. Guarantees on the deployed service’s delay are important as many emerging applications are delay sensitive. Yet, besides some exceptions [23], delay guarantees are not well-studied in the current state-of-the-art. Other solutions propose a delay-aware model [24], but the presented greedy heuristic does not rigorously address the delay guarantee, as the service delay is the optimization objective. Minimizing the service delay often disregards other important optimization goals, such as deployment cost. Authors of [25] study the trade-off between load balancing the compute infrastructure and minimizing the end-to-end service delay, but the solution lacks delay guarantees, as the SFC delay is not considered as an optimization constraint. The state-of-the-art in service placement related problems lacks heuristic solutions, where delay is taken as a constraint to meet the QoS requirements, and minimize for independent objectives.

Our recent work on service placement took the approach of including delay as an optimization constraint with configurable objective function, and implementing it in an extensible heuristic algorithm framework. In our model, delay has been considered as a path-delay, i.e., end-to-end delay, constraint in the service graph. An earlier version focuses on tackling general service graph structures in the greedy VNF placement and routing algorithm [26]. The algorithm framework has been applied to data-plane component orchestration, where the input structures are translated to the general service graph structure, and the heuristic is tuned to the domain-specific needs [27]. Addressing the network operation scenarios, where service requests continuously arrive beyond 0-day deployments, we have extended our framework with hybrid online-offline optimization capability [28]. Most recently, the algorithm has been integrated with our multi-operator 5G proof-of-concept architecture [19], and its advanced features, such as path anti-affinity, has been studied by complexity theory methodologies. Our proof-of-concept implementation is conceptually similar to the previously referred IoT slice orchestration system in [8]. In this paper, our contribution is the extension of our well-established heuristic framework with mathematically founded guarantees for the end-to-end delay constraints. Our results on end-to-end delay and infrastructure abstraction could be applied and integrated with any heuristic algorithm which uses step-by-step placement of the individual service components.

## 3. Base System Model and Optimization Problem

To model a wide range of the previously mentioned dynamic and heterogeneous computation resource types, we introduce four types of substrate nodes arranged in two dimensions. One dimension is the nature of the node’s capacity in terms of flexibility. Substrate nodes with fixed capacity model physical servers, legacy hardware for hosting PNFs and static virtual computation capacities, etc., while elastic substrate nodes model modern pay-as-you-use cloud platforms and auto-scaling capable Virtual Infrastructure Managers (VIMs) such as a Kubernetes host.

The other dimension of infrastructure capability modeling is the level of abstraction represented by the node. A single computation node is represented by the single types, where the placed VNF is directly instantiated, e.g., a single server or a VIM managing a few VMs running on the same or proximate server racks. Finally, aggregate node types hide information of the underlying computation capabilities and show only an abstract view of multiple (possibly distant) VNF hosting possibilities. Except the aggregate elastic category, each infrastructure type is modeled in this paper and our contributions to the domain of infrastructure modeling are summarized by the considered resource types’ abbreviations in Table 1. The resource types are introduced one-by-one incrementally generalizing the system model, while their challenges are studied in detail. In this section we introduce the basic system model only using the single fixed, denoted by (s.f.), as this is the only type generally considered by most of the related VNE, service placement and SFC embedding literature.

### 3.1. System Model and Problem Formulation

We model the substrate network as an undirected, connected graph $S=(V_{S},E_{S})$, where substrate nodes $V_{S}$ represent the compute resources of any type, their interconnections are the edge set $E_{S}\subseteq V_{R}\times V_{R}$. Notations of the problem formulation and for the rest of the paper are summarized in Table 2. The VNF repository is represented by $R^{\left(vnfs\right)}$, which contains all VNFs supported by any parts of the system, used for building services. Keeping aligned with our primary motivation of modeling the characteristics of the infrastructure, we keep the service model simple. We formulate the problem only for the online variant of the service placement problem, restricted to path request graphs. We introduce a general capacity function $c_{S}$, which decides whether a set of VNFs can be placed to a substrate node due to capacity constraints. This function is incrementally extended as we introduce our proposed model for the heterogeneous substrate nodes. For now, the optimization problem is stated with only single fixed (s.f.) substrate nodes, whose capacity is described by the capacity function $c_{S}^{\left(s.f.\right)}$, while the VNF demand mapped to these substrate nodes is described with the corresponding demand function $d_{R}^{\left(s.f.\right)}$. The service deployment’s lifetime $T_{R}$ is considered at the time of component placement decision, as it might affect the incurring costs and the substrate network’s behavior. The placement cost of a VNF is specified by function $p_{V_{R}}$, taking the hosting time length and the mapping as input. Similarly, the routing cost of a request link is specified by $p_{E_{R}}$. If not stated otherwise, the costs $p_{V_{R}}$ and $p_{E_{R}}$ are regarded as input parameters. The node and link mapping functions, V and E, respectively, describe the request graph embedding solution. Link throughput capacities and requirements are not modeled in this paper as we focus on node resource abstraction. Such placement problem variants can be found in our previous work [19,26,27,28] and the literature [20,21].

Formulation 1 shows the service placement optimization problem, where the request graph may only be a path graph, with both ends fixed to substrate nodes from $V_{S}$ as shown in (3). A demonstrative solution for Formulation 1 and some examples for the notations of Table 2 are shown in Figure 2. Constraints (Equation 1) and (2) ensure that all elements of the service are mapped. The location function $L_{R}$ describes placement requirements for each VNF, which might come from operational policies, privacy restrictions, Service Access Point (SAP) matching rules or VNF functionality constraints. The set of feasible paths of a request edge $P_{R}\left(e_{i}\right)$ represents not detailed path constraints like, link capacity and routing constraints. A basic graph embedding requirement is described by (4), which ensures that the source and termination of a request link’s hosting path goes between the hosts of its ends. The general capacity function $c_{S}$ defines for a set of VNFs whether a substrate node has enough capacity to run all functions. Prevention of overloading any substrate node (considering all resource types for extensions) is described by (5). The general capacity function is specified in (6) for single fixed (s.f.) substrate node types, where the demands are integers, simply summed and upper bounded by the integer capacity. This constraint is later extended as we introduce and study the details of other substrate node types. Finally, the objective is to minimize the overall deployment cost which is the total sum of all VNF hosting cost and all routing costs, as shown in (7).

The example on Figure 2 shows concrete values for some parameters. The service chains starting and ending nodes, v0 and v4, respectively, have fixed locations (not subject to optimization) in the substrate graph s0 and s4. VNF v2 can only be located on substrate nodes s5 and s3, and its cost on the latter location is 1$ for the whole service lifetime TR.
**Formulation 1** Variation of the service placement problem studied in this paper.Input: (VS,ES),(VR,ER),cS,cS(s.f.),dR(s.f.),LR,TROutput: V,E
(1)∀vi∈VR:∃V(vi)∈LR(vi)⊆VS              
(2)∀ei∈ER:∃E(ei)∈PR(ei)⊆P(ES)            
(3)v0,vn+1∈VR:V(v0)=s0,andV(vn+1)=sn+1=t         
(4) ∀(vi,vj)∈ER:V(vi)=E(vi,vj).src_nodeandV(vj)=E(vi,vj).dst_node
(5)∀si∈VS:cS({∀vj|V(vj)=si,vj∈VR},si)=1         
(6)     ∀si∈VS,r⊆VR:cS(r,si)=1,if∑v∈rdR(s.f.)(v)≤cS(s.f.)(si)andTS(si)=(s.f.)0,otherwise
(7)subjecttominimizeV,E∑v∈VR∑e∈ERpVR(TR,v,V(v))+pER(TR,e,E(e))      

### 3.2. Heuristic Greedy Backtrack Approach

This section briefs our greedy backtracking algorithm framework for solving service placement for various constraints in our earlier works. Here we state only the general outline of the algorithm to give sufficient background, and a previously solved service placement peculiarities are listed.

Algorithm 1 takes as input the service and substrate graph structures $R=(V_{R},E_{R})$ and $S=(V_{S},E_{S})$, and returns a complete mapping structure of VNFs and their connections V and E, if any feasible is found. An ordered list of unit orchestration steps, i.e., *legs*
$\left(e_{i},v_{i}\right)$, a pair of VNF and adjacent request link, is stored in list $\Psi_{R}$ and generated by function OrderLegsForMapping($V_{R}$, $E_{R}$). In the current case the ordering is simple, as the request graph $R=(V_{R},E_{R})$ is a path graph between a source and a destination. In more complicated applications this function returns the legs in an order so that their source is always placed by the $M\mathrm{AP}O\mathrm{NE}\mathrm{NF}(e_{i},v_{i})$ function, before their own placement decisions. We note that our results on the substrate graph model are general and independent of the service graph structure. Previously published and applied versions of the greedy backtrack approach are capable of handling general service structures [19,26,27,28]. The $M\mathrm{AP}O\mathrm{NE}\mathrm{NF}(e_{i},v_{i})$ function generates hosting candidates and filters them according to the constraints, as shown in Formulation 1. In case such a greedy step is not successful, the functions $G\mathrm{ET}B\mathrm{ACKTRACK}O\mathrm{PTION}(e_{i},v_{i})$ and $U\mathrm{NDO}G\mathrm{REEDY}M\mathrm{APPING}(e_{i},v_{i})$ are responsible to explore other greedy mapping directions and to maintain the resource reservations, constraint states and mapping structures. When a complete mapping structure solving Formulation 1 is found, Algorithm 1 returns the solution. Our earlier work also studies tackling the runtime complexity of such a greedy mapping approach for the NP-hard service placement problem [19,27], these examinations are out of the scope of this paper.

In our earlier work we used this greedy heuristic service placement algorithm framework in various different settings, including general service topology structure [26], data plane resource orchestration [27], hybrid online-offline operational setting [28], and integrated it into a 5G network architecture supporting advanced resource (anti)-affinity constraints [19]. Our greedy heuristic framework has served as basis for several patents submitted by Ericsson, patenting VNF decomposition option selection, elastic substrate capacity management and VNF interconnection constraints (Patents pending).

The versatility of the greedy backtracking approach allows it to be applied in various service graph embedding scenarios, including 5G-enabled IoT prototype architectures independent of our recent work [8]. Our algorithm framework can be easily extended, ideal for early deployment, prototype architecture integration and continuous development.

In this paper, we propose theoretical foundations to the greedy approach by providing guarantee on the embedding result’s end-to-end delay requirement. Subsequently, we present our mathematical analysis on node capacity allocation in a heterogeneous distributed computing environment by refining the substrate graph model.
**Algorithm 1** Overview of the Greedy Backtrack Metaheuristic for VNEReturns a set of complete mapping structures of service graph *S* to resource graph *R*.1: **procedure** Map(S,R) →V,E2:    $\Psi_{R}\leftarrow O\mathrm{RDER}L\mathrm{EGS}F\mathrm{OR}M\mathrm{APPING}\left(V_{R},E_{R}\right)$   ▹ Lists all VNFs and links to be mapped3:    **while**
$\exists e_{i},v_{i}\in \Psi_{R}$
**where**
$∄V(v_{i})$
**or**
$∄E(e_{i})$
**do**4:    **while**
$M\mathrm{AP}O\mathrm{NE}\mathrm{NF}(e_{i},v_{i})$ not successful **do**5:       $e_{i^{\prime }},v_{i^{\prime }}\leftarrow G\mathrm{ET}B\mathrm{ACKTRACK}O\mathrm{PTION}\left(e_{i},v_{i}\right)$6:       $U\mathrm{NDO}G\mathrm{REEDY}M\mathrm{APPING}(e_{i^{\prime }},v_{i^{\prime }})$7:       $e_{i},v_{i}\leftarrow e_{i^{\prime }},v_{i^{\prime }}$                   ▹ Try remapping another *leg*8:    **end while**9:    **end while**10:    **return**
V,E11: **end procedure**

## 4. Result on Path Delay Constraint

This section extends our system and optimization model with latency values, formalizes path delay constraint and we present our result on how to design the step-by-step delay heuristics to approximate end-to-end delay requirements. The notations related to the delay model are shown in Table 3.

We annotate each substrate graph connection by communication delays, and calculate the distances of each substrate node pairs $\forall s_{i},s_{j}\in V_{S}$, denoted by delay distance $d_{s_{i}s_{j}}$. In our general greedy framework of Algorithm 1, the $M\mathrm{AP}O\mathrm{NE}\mathrm{NF}(e_{i},v_{i})$ function can allocate any path $E(e_{i})$ for hosting the request edge $e_{i}\in E_{R}$, which is aligned with path constraints (see $P_{R}\left(e_{i}\right)$). The delay of such feasible path is denoted by $d_{s_{i}s_{j}}^{\left(fp\right)}$. This substrate graph path delay for hosting a request graph edge is alternatively denoted by $d_{v_{j}⇝v_{j+1}}^{\left(E\right)}$, to eliminate the indirection of the node mapping function V from the notation. Having the VNFs indexed in their order on the path request *R* as $V_{R}=\left\{v_{0},\cdots v_{n},v_{n+1}\right\},E_{R}=\left\{\left(v_{0},v_{1}\right),\cdots \left(v_{n},v_{n+1}\right)\right\}$, we extend the original problem definition of Formulation 1 by a path delay requirement.
(8)$$\begin{array}{c} D_{n+1}^{\left(used\right)}=\sum_{\left(v_{i},v_{i+1}\right)\in E_{R}}d_{v_{j}⇝v_{j+1}}^{\left(E\right)}\leq D^{\left(req\right)} \end{array}$$
where $D^{\left(req\right)}$ denotes the maximal allowed path delay of the used route from the beginning to the end of the service graph path.

Assume that the function OrderLegsForMapping($V_{R}$, $E_{R}$) returns the edges of the path request graph in order, starting from the fix substrate node $s_{0}=V\left(v_{0}\right)$, where the chain starts from. The algorithm updates the remaining delay budget $D^{\left(Vv_{i}\right)}$ by subtracting the used path delay $d_{V\left(v_{i-1}\right)V\left(v_{i}\right)}^{\left(fp\right)}$ after the greedy mapping of a leg leading to VNF $v_{i}$. The termination of the request path graph *R* is in substrate node $V\left(v_{n+1}\right)=s_{n+1}$, which is alternatively referred to as *t*. We state our result on delay guarantee in the following theorem.

**Theorem** **1**(End-to-end delay constraint guarantee)**.**
*A greedy heuristic maps a VNF in each step $\forall v_{i}\in \left\{v_{1},\cdots v_{n}\right\}$, where all VNFs until index i are greedily placed, i.e., $\exists V(v_{i})$, and the hosting substrate node and path of leg $\left(e_{i+1},v_{i+1}\right)$ are being selected.*
*During the greedy mapping of a service graph, if for choosing the host of all VNFs $v_{i+1}\in \left\{v_{1},\cdots v_{n}\right\}$,*

*used delay is a fraction of the path delay requirement: $d_{v_{j}⇝v_{j+1}}^{\left(E\right)}\leq \alpha D^{\left(req\right)},\;\frac{1+\epsilon }{n+1}<\alpha \leq 1$,*

*the used substrate node’s distance is bounded: $d_{V\left(v_{i}\right)V\left(v_{i+1}\right)}+d_{V(v_{i+1})t}>B^{\left(dist\right)}$, and $d_{V(v_{i+1})t}\leq d_{s_{0}t}$*

*and the remaining delay requirement is bounded: $D^{\left(V\left(v_{i}\right)\right)}>B^{\left(req\right)}$,*

*then the used total path delay (1+ϵ) approximates the path delay requirement:*
$$\begin{array}{c} D_{n+1}^{\left(used\right)}\leq \left(1+\epsilon \right)D^{\left(req\right)},\;\epsilon \in R^{+}, \end{array}$$
*where*
(9)$$\begin{array}{c} \;\;\;\;\;\;\;\;0<B^{\left(dist\right)}<\frac{n+\alpha }{n+1}D^{\left(req\right)}, \end{array}$$
(10)$$\begin{array}{c} \;\;\;\;\;\;\;\;\;0<B^{\left(req\right)}<\alpha D^{\left(req\right)}+d_{s_{0}t}, \end{array}$$
(11)$$\begin{array}{c} B^{\left(dist\right)}B^{\left(req\right)}\geq \frac{n+\alpha -\epsilon }{n+1}\left(\alpha D^{\left(req\right)}+d_{s_{0}t}\right)D^{\left(req\right)}. \end{array}$$


The proof of the Theorem 1 is presented in Appendix A.

Theorem 1 helps designing a heuristic algorithm’s step-by-step bounds on the latency values to guarantee the (1+ϵ)-approximation of the end-to-end path delay constraint. It is important to note that the algorithm does not need to minimize for delay to comply with the constraint, i.e., shortest path calculation for each request edge might be done for the cost metric, while keeping the theorem’s bounds on the delay. The corresponding subproblem is the restricted shortest path problem, for which efficient algorithms have been proposed [29].

Furthermore, the presented result does not utilize the previously well-studied backtracking capability of the proposed embedding framework. If substrate elements with the desired bounds can not be found, or the (1+ϵ)-violation of the delay constraint needs to be mitigated, undoing some earlier greedy placement steps may solve the issue. Relation between the (1+ϵ) approximation ratio and the required search space for the backtracking might be studied to design a fully polynomial-time approximation scheme (FPTAS) for the problem. A numerical example for applying Theorem 1 is shown in Section 7.

Having demonstrated how advanced constraints, such as end-to-end delay, can be supported by the greedy backtrack framework, now we turn to refining the substrate graph model which provides powerful benefits for the embedding algorithm.

## 5. Result on Resource Aggregation

### 5.1. Revisit the BiS-BiS Model

Network abstractions of various types have proved useful in the past, such as network abstraction map [12], hose model for VPN design [11] and more recently the Big Switch abstraction in SDN [15]. The Big Switch model describes network segments with their end-to-end transport characteristics. As computation and networking are increasingly interleaving thanks to softwarization and virtualization, the Big Switch approach was extended to describe both by the Big Switch-Big Software (BiS-BiS) abstraction [17,18]. Our work refines the BiS-BiS abstraction and provides theoretic foundation to it.

To describe an abstract view of distributed computation sites and to supply valuable information for latency sensitive applications, we propose adding internal aggregate nodes to the BiS-BiS model which describe transport and/or computation resources. Figure 3 shows an abstract view of the lower level infrastructure view of Figure 1. In this example, the CSP chooses to show its network aggregated into two BiS-BiS domains, describing the west and east segments of its infrastructure by CSP BiSBiS1 and CSP BiSBiS2, respectively. The CSP’s three computing locations, Server2, Server3 and Server4 of Figure 1 are aggregated into the CSP Aggr. node of Figure 3. The other non-aggregated nodes, Server1, DataCenter1 and Cluster1 of Figure 1 are represented by CSP Single, DC and OTT Priv nodes, respectively. To simplify the networking structure and to hide some lower level topological information, forwarding nodes of the infrastructure are also aggregated. For instance, the two abstract forwarding nodes of CSP BiSBiS2 in Figure 3 represent the eastern side of the forwarding topology in Figure 1. The edges connecting the internal nodes of a BiS-BiS are annotated by delay values, describing the transport characteristics. Our embedding algorithm proposed in Section 3 is able to operate on a substrate graph, which has such BiS-BiS domains, and is able to map service graphs with end-to-end delay constraints as described in Section 4. The extensions required for the capacity constraints (6) to model the aggregate capacity are presented in this section.

We assume that defining the set of substrate nodes to be aggregated is out-of-scope of the underlying optimization problem, as the clustering might be more influenced by the CSP’s operational policies than rigorous mathematical measures. An extensive survey on clustering algorithms in sensor networks can be found in the literature [30], while general graph clustering is studied in [31].

As it was shown in previous experiments on abstract infrastructure views [19], communicating the bottlenecks is in the interest of both the OTT SP and the CSP to avoid overbooking and under-utilization. To facilitate the effective communication of the bottlenecks, the aggregation must hold an essential property shown in Remark 1.

**Remark** **1**(Essential property of substrate abstraction)**.**
*If a set of VNFs composing the service graph R seems to be hostable on an aggregate substrate node, then the VNF set must be able to be mapped to the underlying hidden network segment.*

Given the clustering of the underlying substrate nodes which are to be aggregated, the central problem in creating the BiS-BiS view is the aggregation of the hidden nodes’ capacities into a single abstract node’s capacity. The aggregating BiS-BiS view must comply with Remark 1 and maximize the shown capabilities of the hidden network segment. In the next section we formulate this central problem as the resource contraction problem, and propose an effective solution to it.

### 5.2. Resource Contraction Problem

To formulate the resource abstraction problem as an optimization we first restate and generalize the capacity allocation subproblem of Formulation 1, characterized by (Equation 1), (5) and (6).

First, the capacity allocation problem should be independent of the request graph instance R=(VR,ER), as the aggregation view should be applicable to any service supported by the system. So we introduce a multiset of the base VNF set (R(vnfs),mR) including not only any set of VNFs but also their repetitions mR. VNFs in an IoT service might appear multiple times, e.g., data processing and noise filtering might need to be done in multiple distributed locations, close to the sensors or end-users. Notations corresponding to this chapter are shown in Table 4. The set of substrate nodes to be aggregated is denoted by VS′⊆VS. Formulation 2 shows the capacity allocation problem generalized to multisets of the possible VNFs and arbitrary subsets of the substrate graph. The equations are formalized for a listing of the request multiset (R(vnfs),mR), where elements with higher multiplicity are repeated accordingly. All repetitions of all VNF vi must be placed somewhere, as expressed by (13). The problem only checks whether an allocation is possible, solving a subproblem of Formulation 1 with (13) and (14) relating to (Equation 1) and (5), (6) respectively.
**Formulation 2** Capacity Allocation Problem (CAP)Input: VS′,(R(vnfs),mR),cS(s.f.),dR(s.f.)Output: True if ∃V, False otherwise
(12)  N:=size((R(vnfs),mR))
(13)      ∀vi∈{v1,v2,⋯vN}:∃V(vi)∈VS′
(14)         ∀si∈VS′:∑vj|V(vj)=si,vj∈{v1,⋯vN}dR(s.f.)(vj)≤cS(s.f.)(si)

The resource contraction problem shown in Formulation 3 builds on the capacity allocation problem. It takes as input a subset of the substrate nodes VS′⊆VS to be aggregated, and returns an aggregate fixed substrate node s¯ with resource type TS(s¯)=(a.f.) and aggregate capacity cS(a.f.)(s¯)=A. Such an example is shown in Figure 3, where the CSP decides to aggregate its single fixed type servers in Figure 1 into a single aggregate substrate node. This is contained in the BiS-BiS created for the western part of the CSP’s infrastructure and is connected to the other parts of the substrate graph through aggregate forwarding nodes shown in Figure 3. The essential property of substrate abstraction of Remark 1 is formalized in (16). For all VNF sets with any multiplicity, if it seems to be placeable on the aggregate node s¯ using the Capacity Allocation Problem (CAP), then it must be possible to allocate it to the hidden nodes. The capacity of the aggregate node shall be maximized to show as accurate view as possible, which is formalized by the objective in (17). The capacity of the aggregate node should never exceed the sum of the hidden nodes’ capacities, formalized in (Equation 15). Note that for evaluating the condition of (16), the Equation (14) simplifies to dem_sum((VR′,mR′))≤cS(s.f.)(s¯)=cS(a.f.)(s¯)=A.
**Formulation 3** Resource contraction problem (RCP)Input: VS′,R(vnfs),cS(s.f.),dR(s.f.)Output: A,s¯,cS(a.f.):{s¯}↦{A},TS(s¯)=(a.f.)
(15)A∈Z+:A≤∑si∈VS′cS(s.f.)(si)    
(16)         ∀VR′⊆R(vnfs),∀(VR′,mR′):IfCAP({s¯},(VR′,mR′),cS(s.f.)(s¯)=A,dR(s.f.))isTruethenCAP(VS′,(VR′,mR′),cS(s.f.),dR(s.f.))isTrue
(17)subjecttomaximizeA∈Z+A          

To this end, if the RCP is solved for a set of single fixed type (s.f.) substrate nodes $V_{S}^{\prime }\subseteq V_{S}$ with capacity function $c_{S}^{\left(s.f.\right)}$ and its accompanying demand function $d_{R}^{\left(s.f.\right)}$, then their contracted view is the aggregate fixed type (a.f.) substrate node $\bar{s}$ with capacity *A*. Note that it is not necessary to define a different demand function for placing VNFs on aggregate fixed nodes, i.e., $d_{R}^{\left(a.f.\right)}=d_{R}^{\left(s.f.\right)}$. Instead of showing the whole set of substrate nodes $V_{S}^{\prime }$, the CSP can choose to publish the aggregate node, which integrates to the resource view of the OTT SPs.

Extending the substrate graph model with the aggregation node $\bar{s}$ specified by the resource contraction problem, we extend Formulation 1 to include aggregate fixed nodes in (6) as shown in (Equation 18):(18)$$\begin{array}{cc} \forall s_{i} & \in V_{S},r\subseteq V_{R}:c_{S}\left(r,s_{i}\right)=\left\{\begin{array}{c} 1,\;\mathrm{if}\;\sum_{v\in r}d_{R}^{\left(s.f.\right)}\left(v\right)\leq c_{S}^{\left(s.f.\right)}\left(s_{i}\right)\;\mathrm{and}\;T_{S}\left(s_{i}\right)=\left(s.f.\right)\\ 1,\;\mathrm{if}\;\sum_{v\in r}d_{R}^{\left(s.f.\right)}\left(v\right)\leq c_{S}^{\left(a.f.\right)}\left(s_{i}\right)\;\mathrm{and}\;T_{S}\left(s_{i}\right)=\left(a.f.\right)\\ 0,\;\mathrm{otherwise} \end{array}\right. \end{array}$$
This describes a heterogeneous view of the infrastructure, while hiding some topology information as desired. The view might be periodically updated as the capacities saturate, to keep the shared information up-to-date and useful, according to the agreement among the providers.

Similar to many assignment problems, the CAP, and thus the RCP as well are NP-hard. The CAP is a subproblem of the notoriously hard VNE problem, and proven to be NP-hard by reducing it to the famous 3-SAT problem [3]. In the following we present an approximation algorithm for solving the RCP, and we analyze its performance.

### 5.3. Approximating Resource Contraction

We propose a solution to efficiently approximate the NP-hard resource contraction problem (RCP) presented in Formulation 3.

The pseudo-code of aprxRCP is shown in Algorithm 2. The VNF $v_{\max}$ with largest demand $d_{\max}$ is selected from the input base VNF set. In lines 6 and 8 the big and small capacity substrate nodes are defined, based on being or not being able to fit $d_{\max}$, defining a 2-partition of the to-be-aggregated nodes $V_{S}^{\prime }$. Note that operating on integer demands and capacities is important to achieve the approximation result. A component of the final aggregation capacity $A_{0}$ is defined by accounting for at least $d_{\max}$ space in each big node $V_{S}^{\prime (big)}$ in line 7. This way, the maximum amount of repetition of the biggest VNF $v_{\max}$ is handled in the input multiset $\left(R^{\left(vnfs\right)},m_{R}\right)$ of CAP while solving the RCP. In addition, this approach ensures that the CAP on the hidden substrate nodes is possible, as a lower limit on the sum of such a capacity allocation is calculated. The function aprxRCP runs recursively by accounting for the highest amount of reservable capacities for the second largest VNF, by removing $v_{\max}$ from the base VNF set $R^{\left(vnfs\right)}$ and executing the whole function only for the small substrate nodes $V_{S}^{\prime (small)}$ in line 10. As the recursively returned aggregation component $A_{1}$ cannot be higher than $d_{\max}$ (otherwise the essential property of Remark 1 might be violated), a result is produced. A termination criterion for the recursion is an empty input VNF set, which results in summing the capacities of the input substrate nodes (recursively remaining nodes with small capacities) in line 3.

In each recursive call, at least one VNF, the current $v_{\max}$, is taken out of the input VNF set, so the recursion is at most $|R^{\left(vnfs\right)}|$ deep. The algorithm performs polynomial number of steps in the input sizes in each recursive call, so the recursion always terminates and its complexity is $O(|R^{\left(vnfs\right)}{|}^{2}|V_{S}^{\prime }|)$. We state the performance guarantee of aprxRCP presented in Algorithm 2 in relation to the optimal solution of RCP.

**Theorem** **2**($\frac{1}{2}$-approximation of the Resource Contraction Problem)**.**
*Given any subset of the substrate graph nodes $V_{S}^{\prime }\subseteq V_{S}$ and base VNF set $R^{\left(vnfs\right)}$ with capacity function $c_{S}^{\left(s.f.\right)}$ and VNF demand function $d_{R}^{\left(s.f.\right)}$, Algorithm 2 $\frac{1}{2}$-approximates the optimal solution of the Resource Contraction Problem of Formulation 3:*
(19)$$\begin{array}{c} \frac{1}{2}OPT\left\{\mathrm{RCP}\left(V_{S}^{\prime },R^{\left(vnfs\right)},c_{S}^{\left(s.f.\right)},d_{R}^{\left(s.f.\right)}\right)\right\}\leq \mathrm{APRX}\mathrm{RCP}\left(V_{S}^{\prime },R^{\left(vnfs\right)},c_{S}^{\left(s.f.\right)},d_{R}^{\left(s.f.\right)}\right) \end{array}$$

The proof of Theorem 2 is presented in Appendix B. A numeric example for Formulation 3 solved by aprxRCP is presented in Section 7.

Supported by Theorem 2, we can efficiently calculate the capacity of an aggregate fixed (a.f.) type substrate node, hiding a set of single fixed (s.f.) type substrate nodes. This enables a CSP to provide a heterogeneous infrastructure view to its tenant OTT SPs, hiding sensitive parts of the topology, while being able to show exact views where it is necessary, as it is demonstrated in Figure 3. The CSP can use aprxRCP to 12-approximate the optimal aggregate capacity to be shown to its tenants’ resource orchestrators. The presented greedy heuristic framework Algorithm 1 in Section 3.2 can be easily extended to respect the multiple types of substrate nodes, because only the capacity calculation function of Formulation 1 needs to be modified. The aggregation scheme also contributes to the scalability of the embedding algorithm executed by the OTT SP, as it decreases the number of substrate nodes in its infrastructure view. As shown in (Equation 18), the capacity interpretation is extended for (a.f.) type nodes and its implementation can be done in the MAPONENF(ei,vi) function.
**Algorithm 2** Resource contraction algorithm solving Formulation 3Input: VS′,R(vnfs),cS(s.f.),dR(s.f.)Output: *A*1: **procedure**
aprx
RCP($V_{S}^{\prime },R^{\left(vnfs\right)},c_{S}^{\left(s.f.\right)},d_{R}^{\left(s.f.\right)}$) →A2:  **if**
$R^{\left(vnfs\right)}=\emptyset$
**then**3:      **return**
cap_sum($V_{S}$′)4:  **else**5:      $v_{\max},d_{\max}\leftarrow \max_{d_{R}^{\left(s.f.\right)}}\left\{R^{\left(vnfs\right)}\right\}$            ▹ Get the VNF with biggest demand  6:      $V_{S}^{\prime (big)}\leftarrow \left\{s_{i}\;|\;s_{i}\in V_{S}^{\prime },\;d_{\max}\leq c_{S}^{\left(s.f.\right)}\left(s_{i}\right)\right\}$    7:      $A_{0}\leftarrow \sum_{s_{i}\in V_{S}^{\prime (big)}}\max\left\{\lfloor \frac{c_{S}^{\left(s.f.\right)}\left(s_{i}\right)}{2}\rfloor +1,\;c_{S}^{\left(s.f.\right)}\left(s_{i}\right)-d_{\max}+1\right\}$  8:      $V_{S}^{\prime (small)}\leftarrow \left\{s_{i}\;|\;s_{i}\in V_{S}^{\prime },d_{\max}>c_{S}^{\left(s.f.\right)}\left(s_{i}\right)\right\}$  9:      **if**
$V_{S}^{\prime (small)}\neq \emptyset$
**then**10:      $A_{1}\leftarrow \mathrm{APRX}\mathrm{RCP}\left(V_{S}^{\prime (small)},R^{\left(vnfs\right)}\setminus \left\{v_{\max}\right\},c_{S}^{\left(s.f.\right)},d_{R}^{\left(s.f.\right)}\right)$  11:      **return**
$A_{0}+\min\left\{d_{\max}-1,\;A_{1}\right\}$12:      **else**13:      **return**
$A_{0}$14:      **end if**15:  **end if**16: **end procedure**

## 6. Results on Elastic Resources

### 6.1. Modeling Resource Elasticity

To capture important characteristics of state-of-the-art computation infrastructure, we model elastic capacity and on-demand billing. Kubernetes is widely used in the industry today, thanks to its active open source community, enabling the rapid development of practical IoT deployments. As it was demonstrated earlier, Kubernetes-based virtualization architectures are a good candidate to meet the extreme requirements of IoT services [8]. We take the widely used Horizontal Pod Autoscaler (HPA) of the Kubernetes container management system [6] as a representative example to model resource elasticity. We propose a mathematical model of Kubernetes HPA to extend our detailed model of the heterogeneous compute and network infrastructure.

Using Kubernetes terminology, we refer to a single instance of a containerized VNF as pod, which is managed by HPA and replicated or terminated in response to changing workload. By default, HPA measures the average of all instantiated pods’ utilization of their allocated CPU capacity, and compares it to a target utilization [7].

Definition 1 lists the HPA parameters and formalizes its operation, which is the base for describing the behavior of a single elastic substrate node $s_{i}\in V_{S}$.

**Definition** **1**(Kubernetes HPA parameters and operation)**.**
*The following parameters are configured by the operator of HPA [7]:*
*1.* $c_{min}$ and $c_{max}$ are the minimal and maximal allowed pod count,*2.* $\hat{u}\in \left[0,1\right]$ is the targeted scaling metric average of all running pods,*3.* $t^{\left(sca\right)}\in R^{+}$ time length of a scaling interval, when HPA evaluates metrics for scaling decisions,*4.* σ∈[0,1] is the scaling tolerance,*5.* $N^{\left(sca\right)}:=\lfloor \frac{T_{R}}{t^{\left(sca\right)}}\rfloor ,T_{R}>t^{\left(sca\right)}$ is the number of scaling intervals during the lifetime $T_{R}$ of the service R.
*$u_{j}\in \left[0,1\right]$ is the measured scaling metric average of all running pods in scaling interval $j\in \{1,2,\cdots N^{\left(sca\right)}\}$. A scaling decision is made in scaling interval j, when*
(20)$$\begin{array}{c} |\frac{u_{j}}{\hat{u}}-1|>\sigma , \end{array}$$
*and the number of pods in interval j+1 recursively yields:*
(21)$$\begin{array}{c} c_{j+1}^{*}=\lceil c_{j}\frac{u_{j}}{\hat{u}}\rceil . \end{array}$$
*After applying the limits, pods are terminated or instantiated to meet the count*
(22)$$\begin{array}{c} c_{j+1}=\mathrm{APPLY}P\mathrm{OD}L\mathrm{IMITS}\left(c_{j+1}^{*}\right)=\left\{\begin{array}{cc} c_{j+1}^{*} & if\;c_{min}\leq c_{j+1}^{*}\leq c_{max}\\ c_{min} & if\;c_{j+1}^{*}<c_{min}\\ c_{max} & if\;c_{j+1}^{*}>c_{max}. \end{array}\right. \end{array}$$


During a given scaling interval, the number of pods (i.e., VNF replicates) processing the incoming task of a VNF $v_{i}$ is fixed. Elementary tasks of a VNF model are for example sensor data processing tasks, video processing requests, HTTP requests for a webserver VNF, flow classification requests for a firewall VNF or incoming images for an image processing VNF, etc. We extend our VNF model by assigning incoming elementary task rate $\lambda_{i}$ and processing rate $\mu_{i}$ to VNF $v_{i}$. Notations for the service and substrate model extensions of this section are gathered in Table 5.

As the default configuration of HPA uses CPU resources as scaling metric, we primarily assume the modeling characteristics based on the nature CPU utilization of computation intensive tasks. We propose to model the pod utilization states using a classic M/M/c queue, where “c” number of pods are processing the VNF’s incoming tasks in parallel. The corresponding assumptions are collected in Proposition 1.

**Proposition** **1**(Model scaling interval as M/M/c queue)**.**
*A scaling interval of the Kubernetes HPA $j\in \{1,2,\cdots N^{\left(sca\right)}\}$ for a single VNF $v_{i}\in V_{R}$ can be described by a classic M/M/c model supposing the following assumptions:*
*1.* the pods are instantiated and terminated instantly,*2.* the arrival process of the VNF’s task is memoryless with constant rate $\lambda =\lambda_{i}$ during the scaling interval,*3.* each pod has identical characteristics, including the VNF’s task processing with an identical, memoryless process with constant rate $\mu =\mu_{i}$ during the scaling interval,*4.* the task queue of VNF $v_{i}$ has infinite length and the task-to-pod assignment takes negligible time, assuming computation intensive tasks and CPU utilization scaling metrics, the pods have two distinct utilization states: busy and idle, with 100% and 0% scaling metric utilization, respectively,*5.* scaling metric $u_{j}$ is the average utilization of all running pods.

Note that the HPA utilization metric can be easily configured and thus the assumptions of Proposition 1 would still hold for the general scaling metric. For instance, if the application is I/O intensive, and HPA is configured to scale based on I/O resource usage, Proposition 1 remains realistic.

We state our analytical results on the M/M/c queue model in Theorem 3.

**Theorem** **3**(Analysis of M/M/c queue)**.**
*If inequality cμ≤λ holds for and M/M/c queue with arrival rate λ and "c" instances of servers with processing rate μ, then the expected values E of the following real random variables exist and can be calculated as follows:*
*Total busy time $\Theta_{c}$: measures the length of time until the first idle moment of an M/M/c system starting from c busy servers.*$$\begin{array}{c} E\left[\Theta_{c}\right]=-\sum_{k=1}^{c}\frac{d}{ds}\eta_{k}\left(0\right) \end{array}$$Weighted total busy time $\Omega_{c}$: measures the area under the time–busy server count chart for an M/M/c system starting from c busy servers until the first idle moment.
$$\begin{array}{c} E\left[\Omega_{c}\right]=-\sum_{k=1}^{c}\frac{d}{ds}\beta_{k}\left(0\right) \end{array}$$
*where $\eta_{k}\left(s\right)$ and $\beta_{k}\left(s\right)$ are the Laplace–Stieltjes Transform of the random variables composing the total busy time $\Theta_{c}$ and weighted total busy time $\Omega_{c}$ respectively. The exact definitions of the random variables and their calculations are shown in Appendix C.*

The proof of Theorem 3 is detailed in Appendix C. We propose to use the expected values $E[\Theta_{c}]$ and $E[\Omega_{c}]$ to estimate the scaling metric of HPA.

**Proposition** **2**(Usage of Theorem 3 to model HPA)**.**
*The measured scaling metric $u_{j}$ of Kubernetes HPA in any scaling interval $j\in \{1,2,\cdots N^{\left(sca\right)}\}$ for a single VNF $v_{i}\in V_{R}$ can be estimated:*
$$\begin{array}{c} u_{j}\approx \tilde{u}=\frac{E[\Omega_{c}]}{cE[\Theta_{c}]} \end{array}$$
*calculated for an M/M/c queue with server count, arrival and departure rates $c=c_{j}$, $\lambda =\lambda_{i}$ and $\mu =\mu_{i}$, respectively.*

**Argumentation.** 
*The momentary utilization of an M/M/c system in any given time–busy server count trajectory (i.e., a concrete realization of the random process) is calculated by the number of busy servers (pods) divided by the number of total servers "c". Thus, the average utilization in an interval t can be calculated by the area below the time–busy pod count chart until t, divided by the maximal possible (rectangular) area ct. As $\Omega_{c}$ measures the area under the trajectory chart and $\Theta_{c}$ measures the time until the idle state, their fraction gives the average utilization. Applying the assumptions of Proposition 1 this metric equals to the scaling metric which would be measured by HPA on the given pod count trajectory. This estimation neglects the dependence of the two random variables and assumes that the probability $P(t>\Theta_{c})$ is negligible, which are realistic assumptions in a practical setting. Theorem 3 states the expected values $E[\Theta_{c}]$ and $E[\Omega_{c}]$, which can be used to estimate the measured scaling metric $u_{j}\approx \tilde{u}$.*


### 6.2. Describing Elastic Resource Behavior

As described in Definition 1, the only non-deterministic part of HPA is the scaling metric measurement $u_{j}$ in scaling interval *j*. Having the estimation $u_{j}\approx \tilde{u}$ as shown in Proposition 2, we can model an elastic node’s behavior in response to various loads.

Algorithm 3 determines a single elastic substrate node $s_{i}$’s behavior (https://github.com/nadamdb/k8s-hpa-modeling) by the trajectory of the varying pod numbers working on a single to-be-placed VNF’s task during the whole service lifetime $T_{R}$. Procedure modelHPAbehavior takes as input a VNF’s intensity characteristics, and a possible elastic host node’s HPA settings. It iteratively uses the subroutine nextHPAPodCount to determine the nature of the scaling decision in each scaling interval during the lifetime of the service. The pod count trajectory function $f^{\left(pod\right)}$ is continuous in its first argument, which tells for each time instance the predicted number of running pods which would process the VNF $v_{i}$’s tasks on substrate location $s_{i}$. If in any scaling interval the pod count is restricted by the maximal number of pods $c_{max}$, the trajectory is invalidated by terminating the algorithm according to the condition in line 7. As shown in line 17, the algorithm has to handle scaling intervals when the M/M/c model’s stability criterion, i.e., a condition of Theorem 3 is not met. A cumulative overload parameter $\rho^{\left(overload\right)}$ is used between iterations, which compensates the estimated utilization metric by adding the volume of the overload. Line 21 gets the scaling metric estimation based on Proposition 2, and the overload parameter is decreased by the compensated amount. The scaling decision and the number of running pods are determined according to the HPA operation shown in Definition 1 and formalized in Algorithm 3 starting from line 24.

Note that Proposition 1 only requires the arrival and processing rates to be constant during the scaling interval. Algorithm 3 is able to handle different rates in each scaling interval, enabling it to follow dynamic changes in the request arrival or changes in the computation circumstances. The presented algorithm has been tuned to real measurements taken on a Kubernetes HPA to properly model the overloaded states, when Theorem 3 cannot be used. The overload parameter ρ(overload) aligns the model’s prediction with experiments. A validative comparison between the estimated pod count trajectory of Algorithm 3 and a measurement on the real pod count trajectory of a Kubernetes cluster managed by HPA is discussed in Section 7.
**Algorithm 3** Pod count trajectory modeling of Kubernetes HPAInput: vi,sj, parameters for sj defined in Definition 1Output: f(pod):[0,TR]×VR×VS↦N∪{0}1: **procedure**
model HPABehavior(vi,λi,μi,sj::[c0,cmin,cmax,u^,t(sca),σ,N(sca)])2:    c(curr)←c03:    ∀t∈[0,t(sca)]:f(pod)(t,vi,si)=c0       ▹ Initialize output HPA trajectory function4:    ρ(overload)←0                     ▹ Cumulative overload parameter5:    **for**
k∈{1,2,⋯N(sca)}
**do**6:    ck∗,ρ(overload)←NEXTHPAPODCOUNT(c(curr),λi,μi,u^,σ,ρ(overload))7:    **if**
ck∗ > cmax
**then**8:     **return**⊥                    ▹ Invalidate trajectory function f(pod)9:    **else**10:     c(curr)←APPLYPODLIMITS(ck∗,cmin,cmax)       ▹ Use HPA pod limits in (Equation 22)11:     ∀t∈(kt(sca),(k+1)t(sca)]:f(pod)(t,vi,si)=c(curr)12:    **end if**13:    **end for**14:    **return**
f(pod)15: **end procedure**16: **procedure**
nextHPAPodCount(c(curr),λi,μi,u^,σ,ρ(overload))17:    **if**
λi>c(curr)∗μi
**then**                  ▹ In this case the M/M/c is instable18:    ρ(overload)←ρ(overload)+λi−c(curr)∗μi19:    u(curr)←1.020:    **else**                               ▹ Use M/M/c based model21:    u(curr)←min(u˜+ρ(overload),1.0)             ▹ Get u˜ from Proposition 222:    ρ(overload)←ρ(overload)−(u(curr)−ρ(overload))23:    **end if** 24:    **if**
|u(curr)u^−1|>σ
**then**                   ▹ Use (Equation 20) with uj=u(curr)  25:    c(next)←⌈c(curr)u(curr)u^⌉             ▹ Use (Equation 21) with (curr) as scaling interval *j*26:    **else**27:    c(next)←c(curr)28:    **end if**29:    **return**
c(next),ρ(overload)30:**end procedure**

Using this model of an single elastic (s.e.) substrate node, we interpret their capacities, the demands of the to-be-placed VNFs and further extend the capacity and demand functions with an additional type:(23)$$\begin{array}{cc} \forall t\in \left[0,T_{R}\right],\forall v_{i} & \in V_{R},\forall s_{j}\in V_{S},T_{S}\left(s_{j}\right)=\left(s.e.\right):\\ d_{R}^{\left(s.e.\right)} & =\mathrm{MODEL}\mathrm{HPAB}\mathrm{EHAVIOR}(v_{i},\lambda_{i},\mu_{i},s_{j}::\left[c_{0},c_{min},c_{max},\hat{u},t^{\left(sca\right)},\sigma ,N^{\left(sca\right)}\right])\\ d_{R}^{\left(s.e.\right)} & \left(t,v_{i},s_{j}\right)=f^{\left(pod\right)}\left(t,v_{i},s_{j}\right) \end{array}$$
(24)$$\begin{array}{c} \forall s_{i}\in V_{S},T_{S}\left(s_{i}\right)=\left(s.e.\right):\;c_{S}^{\left(s.e.\right)}\left(s_{i}\right)=s_{i}::c_{max} \end{array}$$
The demand function of a single VNF, to-be-placed on an elastic node is defined as the output of Algorithm 3, being either the pod count trajectory $f^{\left(pod\right)}$ or the invalid character ⊥. Assuming a strict QoS approach for each VNF, we allow the placement of a set of VNFs, if their sum of demand functions (i.e., their pod count trajectories) would never go above the maximal pod count $c_{max}$ of an elastic node $s_{i}$. Furthermore, if a VNF alone would exceed $c_{max}$, the containing VNF set is considered not placeable.

The above interpretation of elastic resource capacities and demands, defines the admission control function for computation capacity managed by Kubernetes HPA. The effect of the incoming requests are modeled based on the infrastructure’s configuration, and this is used to compare the predicted dynamic demand to the allocated infrastructure capabilities. Further extending the capacity constraint (Equation 18) (which is an extension of (6) of Formulation 1) we get: (25)$$\begin{array}{cc} \forall s_{i} & \in V_{S},r\subseteq V_{R}:c_{S}\left(r,s_{i}\right)=\left\{\begin{array}{c} 1,\;\mathrm{if}\;\sum_{v\in r}d_{R}^{\left(s.f.\right)}\left(v\right)\leq c_{S}^{\left(s.f.\right)}\left(s_{i}\right)\;\mathrm{and}\;T_{S}\left(s_{i}\right)=\left(s.f.\right)\\ 1,\;\mathrm{if}\;\sum_{v\in r}d_{R}^{\left(s.f.\right)}\left(v\right)\leq c_{S}^{\left(a.f.\right)}\left(s_{i}\right)\;\mathrm{and}\;T_{S}\left(s_{i}\right)=\left(a.f.\right)\\ 0,\;\mathrm{if}\;\exists v\in r:d_{R}^{\left(s.e.\right)}\left(t,v,s_{i}\right)=⊥\;\mathrm{and}\;T_{S}\left(s_{i}\right)=\left(s.e.\right)\\ 1,\;\mathrm{if}\;\forall t\in \left[0,T_{R}\right]:\sum_{v\in r}d_{R}^{\left(s.e.\right)}\left(t,v,s_{i}\right)\leq c_{S}^{\left(s.e.\right)}\left(s_{i}\right)\;\mathrm{and}\;T_{S}\left(s_{i}\right)=\left(s.e.\right)\\ 0,\;\mathrm{otherwise} \end{array}\right. \end{array}$$

In addition to modeling the capacity of an elastic node, the defined demand function enables us to refine the cost model of a VNF deployment on elastic substrate. If the cost of running a pod of VNF $v_{i}$ on elastic node $s_{j}$ for one time unit is $p^{\left(pod\right)}\left(v_{i},s_{j}\right)$, then the total cost of placing the VNF here for the whole service lifetime $T_{R}$ can be calculated:
(26)$$\begin{array}{cc} v_{i}\in V_{R},s_{j}\in V_{S},T_{S}\left(s_{j}\right) & =\left(s.e.\right),T_{R}\in R^{+}:\\ & p_{V_{R}}\left(T_{R},v_{i},s_{j}\right)=p^{\left(pod\right)}\left(v_{i},s_{j}\right)\int_{0}^{T_{R}}d_{R}^{\left(s.e.\right)}\left(\tau ,v_{i},s_{j}\right)d\tau \end{array}$$

The elastic substrate node model integrates into our heterogeneous infrastructure model, enabling the CSP to expose its detailed view of its dynamic infrastructure towards an OTT SP if necessary. The OTT SP’s orchestration algorithm can utilize this information to make VNF placement decisions and accurate cost estimations using pod count trajectory prediction. Implementing Algorithm 3 and Equations (Equation 25) and (Equation 26) inside the $M\mathrm{AP}O\mathrm{NE}\mathrm{NF}(e_{i},v_{i})$ function, enables our greedy heuristic framework to serve as the OTT SP’s orchestration algorithm over the CSP’s hybrid detailed-abstract infrastructure resources.

## 7. Practical Exploitation

In this section we summarize how our results could be integrated with the referred 5G prototype systems [8,19], and how they can be used for the example *remote security camera analytics and alerts* IoT use case. We demonstrate the running time of Algorithm 1, and compare it to the times required by other components of our prototype system. Furthermore, we present numeric examples for applications of Theorem 1, the resource contraction problem solved by Algorithm 2 and comparing the elastic resource model of Algorithm 3 to measurement. Figure 4 positions the presented algorithms in a general 5G-enabled IoT framework. Finally, we briefly discuss the challenges of the aggregate elastic resource type.

### 7.1. Result Application Examples

As shown by Figure 4, the core service graph embedding is executed by the OTT SP’s orchestration logic, which must guarantee the end-to-end delay requirements, and be aware of the various substrate node types of the underlying heterogeneous infrastructure. As it was demonstrated, our contribution enables Algorithm 1 to solve and extended version of Formulation 1 including the delay constraint of (Equation 8) and computation capacity constraints in (Equation 25). Our results empower existing and future 5G architecture deployments to realize various IoT services, spanning wide geographic regions over software-controlled, abstract and dynamic infrastructures provided by multiple communication and computation entities.

#### 7.1.1. Real-World Prototype Experiment

We have conducted experiments in a real-world sandbox environment, where the computation nodes managed by Algorithm 1 are distributed across multiple European locations [19]. The proof-of-concept (PoC) 5G architecture prototype integrates Virtual Infrastructure Managers (VIMs), which expose their computation resources for an OTT SP’s orchestrator running Algorithm 1. Comparing our PoC to Figure 4, the role of a CSP is played by European Internet Service Providers, while the computation infrastructure is provided by the participating institutions of the research project. We measured the deployment time of two types of Robot Control services with high reliability constraints, i.e., having the robot’s control logic provided by two independently deployed VNFs. The details of the sandbox environment, prototype implementation and the experiment scenarios are published in our earlier work [19], here we only summarize the relevant findings.

Table 6 shows the experiment results in terms of timing, for both application types. The difference in the services is their communication underlay, which from the service placement algorithm’s perspective means different number of components. In both cases the running time of Algorithm 1 is orders of magnitudes smaller than the deployment time required by the underlying VIMs for starting and configuring the to be instantiated VNFs. In the IP routing case more components of the whole PoC architecture are involved in creating the service, this is the result of the significantly higher overall time.

We conclude that the computation requirement of Algorihtm 2 do not cause any bottlenecks in the PoC, so enriching its features with the elastic resource modeling and infrastructure aggregating of the well-scaling algorithms presented in this paper is possible. Our current paper complements this PoC with firm theoretical results to propose foundations for modeling the infrastructure resources.

#### 7.1.2. Application of Theorem 1

Our result on greedy heuristic bound for end-to-end delay, can be used to design the step-by-step bounds for guaranteed constraint approximation performance. The OTT SP’s orchestration logic can use the conditions of Theorem 1 to make a path-delay-aware decision while choosing the routing path for individual service graph connections. Path options are illustrated by dashed arrows in Figure 4. The following numerical example demonstrates how such bounds can be designed.

The expression for the bounds product $B^{\left(dist\right)}B^{\left(req\right)}$ of Theorem 1 can be rearranged to express the estimation of ϵ:(27)$$\begin{array}{c} \epsilon \geq n+\alpha -\frac{\left(n+1\right)B^{\left(dist\right)}B^{\left(req\right)}}{D^{\left(req\right)}\left(\alpha D^{\left(req\right)}+d_{s_{0}t}\right)}. \end{array}$$
To numerically calculate the example, we take a 10-long service chain as the service graph, i.e., n=10. Taking a permissive approach for designing the bounds (i.e., $B^{\left(dist\right)}$ and $B^{\left(req\right)}$), and preserving their dependency on problem input parameters, we may take the $\frac{\sqrt{3}}{2}$ of their allowed intervals, as shown in Table 7.

Substituting these bounds to Equation (Equation 27) and making the simplifications we get actual bound for the approximation value ϵ, shown in Table 7. Having the bound for ϵ we can choose the value of α from its allowed interval $\alpha =0.9\in (\frac{14}{43},1]$. This enables us to calculate the numeric value for all other algorithm parameters, presented in the last column of Table 7.

Note that the shown (1+ϵ)-approximation does not utilize the backtracking capability of the greedy heuristic, which drastically lowers the end-to-end delay augmentation or meets the requirement. The shown method formalizes the design process of greedy end-to-end heuristics, to tackle the trade-off between computation complexity and constraint violation. This result enables us to control the computational impact of Algorithm 1 in the whole orchestration process.

#### 7.1.3. Resource Contraction Example Using Algorithm 2

To facilitate the VNF placement on the aggregate fixed (a.f.) type substrate nodes, the CSP wishes to aggregate the capacities of its servers, and show an abstract view of the computation capacities to the OTT SP’s orchestration logic. The *CSP Aggr.* compute node of Figure 4 illustrates this scenario. The VNF placement logic uses the (a.f.) case of (Equation 25) to decide whether a VNF can be placed on the aggregate substrate node. The following numerical example shows the $\frac{1}{2}$-optimal solution of the NP-hard problem of Formulation 3 by our proposed approximation algorithm shown in Algorithm 2.

For simplicity we omit the capacity and demand functions from the inputs of the related problems and algorithms, and only list the substrate node capacities, VNF demands in the CAP, RCP and aprxRCP arguments. We take the base VNF set $R^{\left(vnfs\right)}$ with capacity demands Dem={1,2,4} and to-be-aggregated substrate nodes $V_{S}^{\prime }$ with capacities Cap={2,3,5,7,8}. First, we estimate OPT{RCP(Cap,Dem)}, which is obviously smaller than 20, as 5 times the VNF with demand 4 could be allocated on the aggregate node, i.e., CAP({20},{4,4,4,4,4}) is true, but could not be allocated on the hidden nodes, i.e., CAP(Cap,{4,4,4,4,4}) is false. Our proposed resource contraction algorithm APRXRCP(Cap,Dem), shown in Algorithm 2, in its first iteration with $d_{\max}=4$, calculates $A_{0}=2+4+5=11$ for the big capacities {5,7,8}. In the second iteration for the small capacities, it calculates $A_{1}=\mathrm{APRX}\mathrm{RCP}\left(\left\{2,3\right\},\left\{1,2\right\}\right)=3$. So the aggregate capacity $\mathrm{APRX}\mathrm{RCP}\left(Cap,Dem\right)=A_{0}+\min\left\{d_{\max}-1,A_{1}\right\}=14$, which is at least $\frac{14}{19}$-approximation of the optimal solution.

In conclusion, the CSP can efficiently show its resources meeting the essential property for abstraction, as described in Remark 1, which view can be quickly generated.

#### 7.1.4. Comparison of Algorithm 3 to Measurement

Finally, our results on modeling Kubernetes HPA, the OTT SP, running Algorithm 1, can estimate the trajectory of VNF replication states for the IoT service’s lifetime in response to user traffic. The trajectory estimation is presented in Algorithm 3 and should be implemented in the OTT SP’s orchestration logic, making placement decisions based on (Equation 25). This is illustrated by the placements of *Extract DB* VNF and *Webserver* VNF to single elastic (s.e.) substrate nodes on Figure 4.

Figure 5 shows the predicted pod count trajectory and its comparison to the behavior of a real Kubernetes cluster deployment managed by HPA. The experiment uses a webserver VNF packaged in a pod, receiving independent, identically and exponentially distributed user requests with constant arrival ($\lambda_{i}=10\;\frac{request}{min}$) and processing intensity ($\mu_{i}=1\;\frac{request}{min}$) for an hour.

The HPA’s scaling interval $t^{\left(sca\right)}$ is set to 1 min, with targeted scaling metric $\hat{u}=0.5$, scaling tolerance σ=0.1 and pod limits $c_{min}=1$, $c_{max}=20$. The model utilizes the calculated arrival and processing rate of the webserver VNF during the current scaling interval. So that, not only a fixed pair of arrival rate $\lambda_{i}$ and processing rate $\mu_{i}$ is given (as shown in Algorithm 3), but the realization of the randomized processes is used to calculate the local rates. This approach slightly generalizes the algorithm to provide a reasonable comparison to the measurement, as the Kubernetes HPA is not aware of the exact rates used for generating the arrival and processing times. These estimated rates are used as the input for the nextHPAPodCount() function to predict the utilization metric, and thus predict the number of active pods for the next scaling interval. The experiment starts with a single running pod, when both the model and the measurement initially over-provisions with very similar extent and timing. Both trajectories follow the slight oscillation around the equilibrium pod count of 11–13, resulting in at most −/+15% deviation.

Using this model the OTT SP can make a well-informed decision for deploying its IoT service over the CSP’s elastic and heterogeneous infrastructure.

### 7.2. Aggregate Elastic Substrate Node Type

Designing a model for aggregate elastic (a.e.) infrastructure node types is extremely challenging, as modeling elasticity requires various parameters describing the dynamic behavior, while aggregation abstracts the lower level infrastructure details. For instance, aggregating the view of multiple Kubernetes HPA nodes using the resource contraction problem (RCP) would require allocating a fix capacity to each of the to-be-hidden clusters. This goes against preserving the dynamic nature of the underlying nodes. On the other hand, describing the aggregate elastic node’s capabilities using well-studied queuing systems, such as the M/M/c queue, would require crude simplifications of infrastructure characteristics.

This is a possible future research direction in infrastructure modeling as dynamicity increasingly gains momentum in cloud computing, while network management of edge resources is already challenging, calling for higher computing and communication resource abstractions.

## 8. Conclusions

We have identified a gap between the rapid evolution of virtualization-based infrastructure technologies and the essentially unchanged substrate models of service placement algorithms. Distributed computing solutions, such as Mobile Edge Computing and 5G utilize increasingly capable infrastructure technologies for addressing quickly changing user demands, and scaling up to the immense sizes of IoT networks. On the contrary, the service component placement algorithms, managing service deployment on top of these architectures, only consider fixed substrate capacities.

Primarily, we have presented our resource aggregation model to support the scalability of network management algorithms, and help preserving sensitive infrastructure information during inter-provider control-plane communication. Secondly, we have used our classic queuing theory results to build an elastic resource model for the auto-scaling mechanism of the wide-spread container management system, Kubernetes. Our results on infrastructure modeling empower existing 5G prototype architectures to place delay-constrained IoT service components on highly dynamic and abstract resource views. In addition, we demonstrated how scalable service placement algorithms can meet path delay requirements by making greedy decisions, while optimizing for independent metrics. Our results integrate with placement algorithms using elementary service-to-resource mapping decisions, enabling their easy deployments already in the proof-of-concept phase. All of our results are complemented with rigorous mathematical theorems whose proofs have been presented in detail.

Finally, we have showcased the practical applicability of our theorems and algorithms to demonstrate what roles our results play for realizing the *remote security camera analytics and alerts* IoT application across wide geographic areas. The infrastructure resources provided by multiple CSPs and CPs are used by customers facing OTT SPs, who provide IoT services. All players of these complex business relations benefit from higher resource abstraction, accurate modeling of dynamic resources and efficient service placement algorithms.

The computation and communication infrastructure for realizing various IoT applications is becoming increasingly capable to keep up with the user requirements for high service quality. In emerging modern networks, the management algorithms not only have to handle communication resources, but the coordinated management of computation needs to be blended in. The contributions of this paper are believed to be essential for effective abstract resource information exchange and detailed modeling of elastic resources. We hope that our results could serve as a basis for such communication protocol definitions and standardized data models for multi-domain, heterogeneous 5G infrastructures.

## Figures and Tables

**Figure 1 sensors-20-03830-f001:**
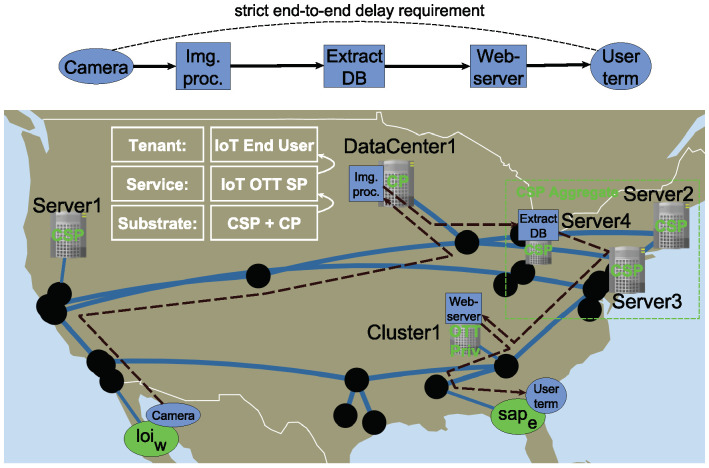
Demonstration of a wide area network with heterogeneous computation components with multiple stakeholders: Communication Service Provider (CSP) and Cloud Provider (CP) selling infrastructure to Over-the-Top Service Provider (OTT SP), who provide remote security camera analytics and alerts Internet of Things (IoT) services to its tenants.

**Figure 2 sensors-20-03830-f002:**
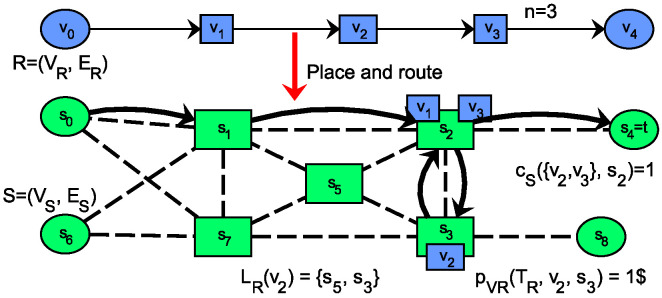
Demonstration of placing a path request graph, meeting the constraints of Formulation 1.

**Figure 3 sensors-20-03830-f003:**
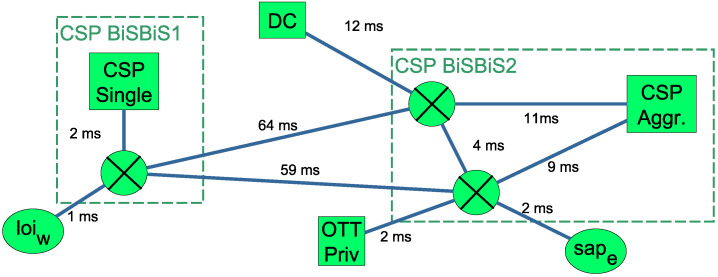
Demonstration of modeling an abstract view of the heterogeneous computation and network infrastructure (rented from the CSP and CP) of Figure 1. This figure shows the topological information which is used by the OTT SP’s orchestration logic to deploy services across the abstract multi-domain view.

**Figure 4 sensors-20-03830-f004:**
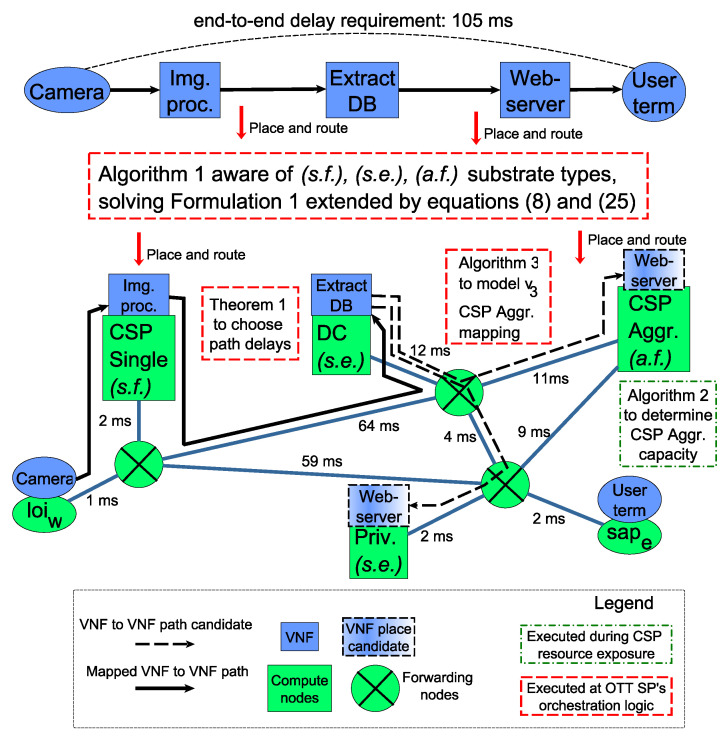
Illustration of applying the presented results in a 5G prototype architecture for the remote security camera analytics and alerts IoT application.

**Figure 5 sensors-20-03830-f005:**
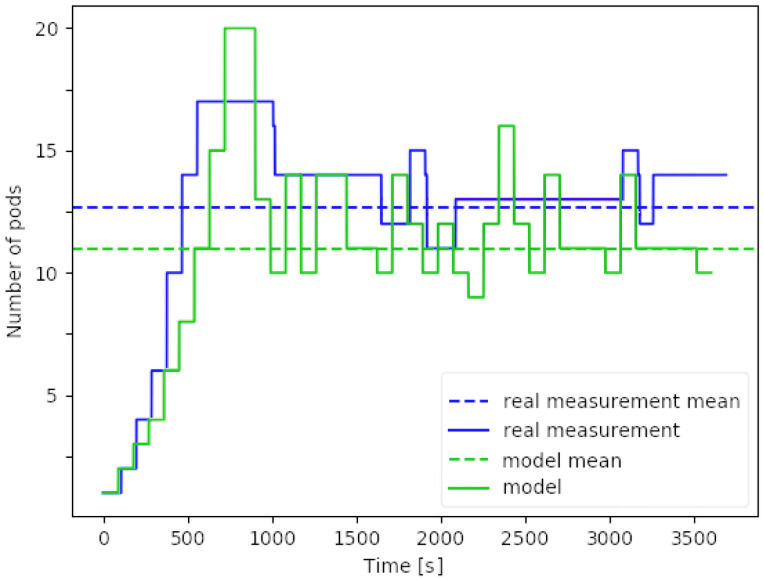
Validative example of single elastic resource model by Algorithm 3 compared to a real Kubernetes Horizontal Pod Autoscaler (HPA) pod count trajectory.

**Table 1 sensors-20-03830-t001:** The modeled substrate node types in the dimensions of flexibility and abstraction, summarizing our contribution to the state-of-the-art (StoA) and the studied substrate node type abbreviations.

	Fixed	Elastic
single	(s.f.)	(s.e.)
	Considered in majority of SotA	Our contribution based on M/M/c results
aggregate	(a.f.)	(a.e.)
	Our contribution based on BiS-BiS model	Briefly discussed

**Table 2 sensors-20-03830-t002:** Notations used in embedding algorithm description.

Notation	Description
$S=(V_{S},E_{S})$	Substrate graph structure for hosting the request *R*
$R^{\left(vnfs\right)}$	Base set of all VNFs used to build any type of service
$R=\left(V_{R},E_{R}\right),\;V_{R}\subseteq R^{\left(vnfs\right)}$	Request path graph of *n* VNFs connecting $s_{0},t\in V_{S}$
P(Q)	Power set, i.e., all possible subsets, of set Q
$L_{R}:V_{R}\mapsto P\left(V_{S}\right)$	VNF placement constraints
$P_{R}\left(e_{i}\right)\subseteq P\left(E_{S}\right),e_{i}\in E_{R}$	Set of feasible paths for request edge $e_{i}$
$c_{S}:P\left(R^{\left(vnfs\right)}\right)\times V_{S}\mapsto \left\{0,1\right\}$	General capacity function deciding hosting capabilities
$T_{S}:V_{S}\mapsto \left\{\left(s.f.\right),\left(s.e.\right),\left(a.f.\right),\left(a.e.\right)\right\}$	Substrate node type of a substrate node
$c_{S}^{\left(s.f.\right)}:V_{S}\mapsto Z^{+}$	Capacity function of single fixed substrate node types
$d_{R}^{\left(s.f.\right)}:R^{\left(vnfs\right)}\mapsto Z^{+}$	Demand of a VNF if mapped to a single fixed substrate node type
$T_{R}\in R^{+}$	Number of time units the service is expected to run
$p_{V_{R}}:R^{+}\times R^{\left(vnfs\right)}\times V_{S}\mapsto R^{+}$	Mapping cost of a VNF to a substrate node for a time interval
$p_{E_{R}}:R^{+}\times E_{R}\times P\left(E_{S}\right)\mapsto R^{+}$	Mapping cost of a request edge to a substrate path for a time interval
$V:V_{R}\mapsto V_{S}$	Hosting substrate node of request node (VNF)
$E:E_{R}\mapsto P\left(E_{S}\right)$	Hosting substrate path of request edge
$e_{i},v_{i}\in E_{S}\times V_{S}=\Psi_{R}$	Leg, the unit orchestration step

**Table 3 sensors-20-03830-t003:** Notations for the end-to-end latency results.

Notation	Description
$d_{s_{i}s_{j}},\;s_{i},s_{j}\in V_{S}$	Delay distance of two substrate nodes
$d_{s_{i}s_{j}}^{\left(fp\right)},\;s_{i},s_{j}\in V_{S}$	Delay of a feasible path between two substrate nodes.
$V\left(v_{0}\right)=s_{0},V\left(v_{n+1}\right)=s_{n+1}=t$	Source and target of the path request graph.
$D^{\left(req\right)}$, $D_{n+1}^{\left(used\right)}$	Required and used path delay on VNF path $V_{R}=\left\{v_{0},\cdots v_{n},v_{n+1}\right\}$.
$d_{v_{j}⇝v_{j+1}}^{\left(E\right)}=d_{V\left(v_{j}\right)V\left(v_{j+1}\right)}^{\left(fp\right)}$	Delay of the hosting substrate path between $V(v_{j})$ and $V(v_{j+1})$
$D^{\left(V\left(v_{i}\right)\right)}=D^{\left(req\right)}-\sum_{j=0}^{i-1}d_{v_{j}⇝v_{j+1}}^{\left(E\right)}$	Remaining delay budget while all VNFs until $v_{i}$ are mapped.

**Table 4 sensors-20-03830-t004:** Notations for modeling the Big Switch-Big Software (BiS-BiS) nodes.

Notation	Description
$\left(Q,m_{Q}\right),\;m_{Q}:Q\mapsto N\cup \left\{0\right\}$	Multiset of set *Q*, with multiplicity function $m_{Q}$
$size\left(\left(Q,m_{Q}\right)\right)=\sum_{q\in Q}m_{Q}\left(q\right)$	Size of a multiset, i.e., including item multiplicity
$V_{S}^{\prime }\subseteq V_{S}$	Subset of the substrate nodes for allocation/aggregation
CAP($V_{S}$’, (Q, $m_{Q}$), $c_{S}^{\left(s.f.\right)}$, $d_{R}^{\left(s.f.\right)}$)	Shorthand of solving the problem of Formulation 2
$cap\_sum\left(V_{S}^{\prime }\right)=\sum_{s_{i}\in V_{S}^{\prime }}c_{S}^{\left(s.f.\right)}\left(s_{i}\right)$	Sum of capacities of a substrate node set $V_{S}$’
$dem\_sum\left(\left(V_{R}^{\prime },m_{R^{\prime }}\right)\right)=\sum_{v_{i}\in V_{R}^{\prime }}d_{R}^{\left(s.f.\right)}\left(v_{i}\right)m_{R^{\prime }}\left(v_{i}\right)$	Sum of demands of VNF multiset $\left(V_{R}{}^{\prime },m_{R{}^{\prime }}\right)$
$c_{S}^{\left(a.f.\right)}:V_{S}\mapsto Z^{+}$	Capacity function of aggregate fixed type substrate nodes

**Table 5 sensors-20-03830-t005:** Notations for modeling the elasticity of substrate nodes.

Notation	Description
$\lambda_{i}\;\forall v_{i}\in V_{R}$	Incoming elementary task rate of VNF $v_{i}$
$\mu_{i}\;\forall v_{i}\in V_{R}$	Processing rate of elementary tasks of VNF $v_{i}$ running in a single instance
$s_{i}::\left[x,y\right],\;s_{i}\in V_{S}$	Local parameters *x* and *y* for substrate node $s_{i}$, e.g., parameters of Definition 1
$c_{j},c_{j}^{*},c_{min},c_{max},c^{\left(curr\right)}$	Pod numbers in various situations
$u_{j},\hat{u},u^{\left(curr\right)},\tilde{u}$	Scaling metric in various situations
$c_{S}^{\left(s.e.\right)}:V_{S}\mapsto Z^{+}$	Capacity function of single elastic substrate node type.
$d_{R}^{\left(s.e.\right)}:R^{+}\times V_{R}\times V_{S}$	Demand function of a VNF to-be-placed on (s.e.) type substrate node
↦N∪{0}	

**Table 6 sensors-20-03830-t006:** Sandbox experiments of our prototype implementation of Algorithm 1 [19].

	Robot Control (SDN Routing)	Robot Control (IP Routing)
Running time of Algorithm 1	0.25 s	0.32 s
VNF deployment at the VIMs	110.23 s	217.96 s
Overall time	112.67 s	219.2 s

**Table 7 sensors-20-03830-t007:** Numerical example for applying Theorem 1 for greedy algorithm design.

Algorithm	Chosen Formula	Numerical Value
Parameter		(n=10,α=0.9)
$B^{\left(dist\right)}$	$\frac{\sqrt{3}}{2}\frac{n+\alpha }{n+1}D^{\left(req\right)}$	0.85$D^{\left(req\right)}$
$B^{\left(req\right)}$	$\frac{\sqrt{3}}{2}\left(\alpha D^{\left(req\right)}+d_{s_{0}t}\right)$	0.78$D^{\left(req\right)}$+ 0.87$d_{s_{0}t}$
ϵ	$\frac{1}{4}\left(n+\alpha \right)$	2.72

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
