# Peer review of "Advanced Computation Capacity Modeling for Delay-Constrained Placement of IoT Services"

_sensors, 2020, doi:10.3390/s20143830_

Round 1

Reviewer 1 Report

No comments.

Reviewer 2 Report

I have read the submission and the rebuttal letter written by the authors. Overall, I am satisfied with the clarifications and changes made in the manuscript, therefore I have no further reasons to object the manuscript acceptance.

This manuscript is a resubmission of an earlier submission. The following is a list of the peer review reports and author responses from that submission.

Round 1

Reviewer 1 Report

The authors consider the context of the 5G technology with IoT and VNE to propose a methodology for aggregating capacities in the field of 5G-enabled IoT service placement, considering Kubernetes containers. A VNE heuristic is proposed to meet the delay constraints. Therefore, the problem is a hot topic and interesting.

The paper is well written, presented and organized.

In table 3, the notation for the remaining delay budget while all VNFs until vi are mapped is a bit confusing.

Figure 8 requires a clarification.

In Chapter 6, you must clarify why you consider a M/M/c queue for the scaling interval of the HPA of the Kubernetes for a single VNF. Is this realistic ? (for instance, the constant rate lambda, or the 2 utilization states).

Section 7.1 must be better clarified. How the computational time required to solve the algorithm can impact in practical deployments ?

An example with numbers from a real scenario (data center, etc) will be very useful to understand the impact of the proposal.

The conclusions section must be extended in order to be more precise to clarify the findings provided in the paper, highlighting the practical significance of the Theorems in the paper.

Author Response

Dear reviewer,

We appreciate your valuable comments to make our paper more understandable, and thus your significant contribution to reaching the desired impact of our results.
Please find all modifications made to the manuscript using the "\change{}{}" custom LaTeX command. This strikes through the deleted text and displays additions in blue.
Please find below our replies to your reviewer comments in their order.

"In table 3, the notation for the remaining delay budget while all VNFs until vi are mapped is a bit confusing.":

- We have added an intermediate variable to eliminate the indirection of the node map function from the notation.

"Figure 8 requires a clarification.":

- We assume the reviewer means Figure 3, as there is no Figure 8 in the manuscript. We have added ellaboration in the caption, and explained how this figure abstracts the view of Figure 1 in more detail.

"In Chapter 6, you must clarify why you consider a M/M/c queue for the scaling interval of the HPA of the Kubernetes for a single VNF. Is this realistic ? (for instance, the constant rate lambda, or the 2 utilization states).":

- Assuming 0% and 100% utilization states is a good model for computation intensive tasks, because in these cases the CPU can be efficiently utilized if there is task to do.
On the other hand if the task is I/O intensive, Kubernetes HPA can be configured to use I/O resource utilization as scaling metric.
It is important to note that, our solution does not directly model the CPU utilization, but the pod utilization metric configured for the HPA (see line 480).
So if the HPA configuration and the model are properly aligned then this assumption is close to reality.
All in all, we find this model a realistic assumption with the appropriate Kubernetes HPA configuration. We show a validative example in Figure 5 to demonstrate how close the model's predictions are to real measurements.

"Section 7.1 must be better clarified. How the computational time required to solve the algorithm can impact in practical deployments ?
An example with numbers from a real scenario (data center, etc) will be very useful to understand the impact of the proposal.":

- We have included in the paper one of our earlier experiement results published in [19] ([20] in the original manuscript) to showcase the timing relations of our 5G architecture prototype implementation.
Section 7.1.1 shows that the running time network management algorithms is much lower than the time required for infrastructure configuration.
Thus we conclude, adding features to the management architecture does not create a bottleneck in the service deployment procedure.

"The conclusions section must be extended in order to be more precise to clarify the findings provided in the paper, highlighting the practical significance of the Theorems in the paper.":

- We have added an extension to the conclusion to summarize the relation to the IoT example use case, considered in the beginning of the paper.
Furthermore we summarized our results' impact on the involved players of the business context.

Best regards,
Balazs Nemeth,
MTA-BME Network Softwarization Research Group
Budapest University of Technology and Economics,
Budapest, Hungary

Reviewer 2 Report

The authors approach service placement in 5G-enabled IoT environments. The authors build upon the literature of virtual network embedding to formalize the placement of services given delay constraints and cost minimization objectives. The authors provide a set of mathematical definitions, lemmata, and theorems to provide theoretical support to their models and take advantage of greedy approaches to find a solution for the model.

The paper is overly dense and difficult to follow. More importantly, the scope of the research is confusing and unclear throughout the manuscript. The authors position their manuscript as addressing the problem of service placement in 5G-enabled IoT environments. Still, they do not consider typical constraints found in IoT environments, like battery, processing, and data communication capabilities. In Sec. 2.1 the authors argue that 5G technology is capable of meeting IoT communication requirements. Still, this claim seems to largely ignore battery constrained devices and those using LPWAN (e.g. LoRa) for data communication.

The authors' formalization and algorithms also seem to tackle the problem from a perspective not considered in previous investigations. For example, Lewin-Eytan et al. [B] approached the problem using solvers to pursue near optimal placement of services, considering a set of requests and an infrastructure with a fixed capacity. In contrast, the authors of the submitted manuscript only consider a greedy approach that basically reduces the problem to finding a feasible but not (near) optimal placement.

There also some aspects not (clearly) addressed in the paper. For example, are the authors working on an online or offline version of the problem? This should be an explicit goal of the investigation, given the authors' intent to keep up with the dynamicity of communication networks required by IoT.

The authors also enumerate as a core contribution of their manuscript an admission control for the Kubernetes container
69 management system. However, admission control is not mentioned at all later in the paper, when discussing the applicability of their solution with Kubernetes HPA. The authors do not clarify either which aspects ought to be considered regarding admission control (e.g., higher-level management or security policies). One could assume that admission control would refer to the possibility of accommodating incoming requests in the infrastructure given the current state of the infrastructure, available resources, and service requirements. That would require approaching an online version of the VNE placement problem, which does not seem to be the case.

The authors mention in Sec. 2.3 that the service component placement problem related to SDN and NFV is well studied as the VNE problem. This is incorrect. VNE refers to the instantiation of virtual networks on top of a substrate infrastructure, whereas the VNFPC (virtual network function placement and chaining) refers to the use of SDN to place VNF instances in network infrastructure, and steer traffic between those instances by taking advantage of SDN capabilities. There have been several authors who approached the VNFPC problem under various scenarios and constraints, considering online/offline and (un)capacitated versions. Moens et al. [A] were the first to approach VNFPC by approaching it as a formalization problem, Lewin-Eytan et al. [B] approached uncapacitated and capacitated versions of the problem, and Luizelli et al. [C] highlighted the commonalities and differences between VNE and VNFPC, proving that VNFPC belongs to class NP-complete.

Some minor comments:

- Consider citing a peer-reviewed and more recent survey on the VNF placement problem [D]

- Formulation 1 (F1) has a parallel with many investigations that approach the placement of service function chains in a given infrastructure, with the difference that F1 only considers the placement of a single SFC. The backtracking-based greedy heuristic described to solve it is a naive one, that hardly scales to real-world scenarios. The extension presented in Sec. 4 is again largely approached in existing work (

- For completeness, please describe your constraint sets in Formulation 1
- The formulation of the theorems presented in the paper is overly complex. For example, the first part of Theorem 1, before the logical implication, could be left off the theorem statement. Theorem 2 includes a whole algorithm that introduces a complexity beyond reasonable. Theorem 3

- pg 3. line 103: points >>while<< meeting the strict delay requirement

- pg 3. line 103: The CSP provides computation capabilities by its distributed cloud? You probably might mean CP here.

- Ref [6] is not available to readers. Please publish the resource in an open access repository like arXiv, or remove the reference.

[A] H. Moens, F. De Turck. Vnf-p: a model for efficient placement of virtualized network functions. Proceedings of CNSM 2014, pp. 418-423

[B] L. Lewin-Eytan, J. Naor, R. Cohen, D. Raz. Near optimal placement of virtual network functions. Proceedings of INFOCOM 2015, IEEE, New York, NY, USA (2015), pp. 1346-1354

[C] Luizelli, Marcelo Caggiani, et al. "A fix-and-optimize approach for efficient and large scale virtual network function placement and chaining." Computer Communications 102 (2017): 67-77.

[D] Laghrissi, A. and Taleb, T., 2018. A survey on the placement of virtual resources and virtual network functions. IEEE Communications Surveys & Tutorials, 21(2), pp.1409-1434.

Author Response

Dear reviewer,

We appreciate your valuable comments to make our paper more understandable, and thus your significant contribution to reaching the desired impact of our results.
Please find all modifications made to the manuscript using the "\change{}{}" custom LaTeX command. This strikes through the deleted text and displays additions in blue.
Please find below our replies to your reviewer comments in their order.

"The paper is overly dense and difficult to follow. More importantly, the scope of the research is confusing and unclear throughout the manuscript. The authors position their manuscript as addressing the problem of service placement in 5G-enabled IoT environments. Still, they do not consider typical constraints found in IoT environments, like battery, processing, and data communication capabilities.":

- We consider modeling the backend of IoT services running on the distributed infrastructure of edge-cloud computing.
Our results propose a detailed infrastructure model, describing the modern capabilities of autoscaling and resource abstraction.
These models can be used by various types of service placement algorithms to make well informed component placement decisions.
We use the service placement algorithm presented in Algorithm 2 to put our infrastructure models in context.
This algorithm can be replaced with other placement algorithms supporting typical IoT requirements such as processing needs, battery models, data communication capacities and many more.
Furthermore, our problem definition is general enough to accommodate additional requirements. Table 2 shows the general notations for VNF placement constraints and feasible path sets for each service graph edge.
We believe this special issue is the right forum for our work to reach its audience and deliver impact.

"The authors' formalization and algorithms also seem to tackle the problem from a perspective not considered in previous investigations. For example, Lewin-Eytan et al. [B] approached the problem using solvers to pursue near optimal placement of services, considering a set of requests and an infrastructure with a fixed capacity. In contrast, the authors of the submitted manuscript only consider a greedy approach that basically reduces the problem to finding a feasible but not (near) optimal placement.":

- We formalize a simplified version of our greedy service graph placement algorithm to provide the necessary background for our proposed infrastructure models.
We try to keep the service placement problem formulation fairly simple (e.g. disregarding bandwidth constraints, considering only path graphs) and general (formulate an extensible capacity function in equations (5) and (6)) to be able to focus on the substrate models.
Solver-based approaches have the main limitation of utilizing static infrastructure models and exhibiting bad scalability for the immense sizes of IoT networks.
We have modified the related work section to reflect these statements.

"There also some aspects not (clearly) addressed in the paper. For example, are the authors working on an online or offline version of the problem? [..]":

- We consider an online version of the service placement problem for Algorithm 2. Clarification for this has been added.
The rest of our results is independent of whether the placement problem is online or offline, as both of these settings require accurate infrastructure models, describing modern cloud capabilities.

"The authors also enumerate as a core contribution of their manuscript an admission control for the Kubernetes container management system. However, admission control is not mentioned at all later in the paper, when discussing the applicability of their solution with Kubernetes HPA. The authors do not clarify either which aspects ought to be considered regarding admission control (e.g., higher-level management or security policies). One could assume that admission control would refer to the possibility of accommodating incoming requests in the infrastructure given the current state of the infrastructure, available resources, and service requirements. That would require approaching an online version of the VNE placement problem, which does not seem to be the case.":

- The interpretation of our results on Kubernetes HPA can be used to predict how a specific set of VNFs placed on a Kubernetes cluster would influence the state of the infrastructure.
This is detailed in section 6.2, but haven't been clearly explained in the original manuscript. We have clarified it in the updated manuscript by explicitly pointing out the admission control function.

"The authors mention in Sec. 2.3 that the service component placement problem related to SDN and NFV is well studied as the VNE problem. This is incorrect. VNE refers to the instantiation of virtual networks on top of a substrate infrastructure, whereas the VNFPC (virtual network function placement and chaining) refers to the use of SDN to place VNF instances in network infrastructure, and steer traffic between those instances by taking advantage of SDN capabilities. [...] Moens et al. [A] were the first to approach VNFPC by approaching it as a formalization problem, [...]":

- We have removed the incorrect sentence which states that service placement is studied as VNE in the state-of-the-art.
We have rewritten the section 2.3 to correctly explain the relationship of the service placement problem to other related research problems such as VNF Placement, SFC Resource Allocation and Virtual Network Embedding.
We argue that these problems are fundamentally similar, with many various constraints which worth individual studies. This argument is supported by the survey in [21], which studies the relationships of these problems, and categorizes their mutually useful results.
We have added a reference to Moens et al. [A]., it is [22] in the updated manuscript.
Other parts of the paper have been also adapted to clarify this.

"Consider citing a peer-reviewed and more recent survey on the VNF placement problem [D]":

- This reference is included in the paper as [5].

"Formulation 1 (F1) has a parallel with many investigations that approach the placement of service function chains in a given infrastructure, with the difference that F1 only considers the placement of a single SFC. The backtracking-based greedy heuristic described to solve it is a naive one, that hardly scales to real-world scenarios.":

- As mentioned earlier, we aim for a simple statement of the service placement problem, as this is sufficient to provide background for our substrate models.
Greedy backtracking based approaches heavily rely on the heuristic guiding the search, we have studied this problem in our previous works.
Please find a demonstrative experiment of the PoC running times in section 7.1.1 taken from our previous work in [19].

"The extension presented in Sec. 4 is again largely approached in existing work (":

- Our contribution in section 4 is not the fact itself of including the path delay in the service placement problem, but breaking it down to link-wise delay bounds while guaranteeing the approximation of path delay constraint.
In addition to this, in many solutions of the state-of-the-art, path delay or end-to-end delay requirements are handled as minimization objectives, which limits the optimization algorithm's capabilities to minimize for deployment cost.
In the remainder of our paper, we argue that this capability can be integrated with service placement algorithms, benefiting from our proposed infrastructure models.

"The formulation of the theorems presented in the paper is overly complex. For example, the first part of Theorem 1, before the logical implication, could be left off the theorem statement. Theorem 2 includes a whole algorithm that introduces a complexity beyond reasonable.":

- The logical implications between the first sentence of Theorem 1 and the rest of the theorem is clarified.
The complexity of Algorithm 5 (included in Theorem 2) has a polynomial running time complexity with low orders, as shown in lines 448-452.
Algorithm 5 1/2-approximates the RCP problem, as this is related to the NP-hard assignment problem class.

Modifications in line 103:

- Corrected and clarified.

" Ref [6] is not available to readers. Please publish the resource in an open access repository like arXiv, or remove the reference.":

- [6] is removed.

Best regards,
Balazs Nemeth,
MTA-BME Network Softwarization Research Group
Budapest University of Technology and Economics,
Budapest, Hungary